Host generalists dominate fungal communities associated with alpine knotweed roots: a study of Sebacinales

http://orcid.org/0000-0002-4453-4173 Schön Max Emil 1 2 max-emil.schoen@mr.mpg.de
Abarenkov Kessy 3
Garnica Sigisfredo 4 sigisfredo.garnica@uach.cl
1 Department of Biomolecular Mechanisms, Max Planck Institute for Medical Research , Heidelberg , Germany
2 Department of Cell and Molecular Biology, Uppsala University , Uppsala , Sweden
3 Natural History Museum, University of Tartu , Tartu , Estonia
4 Instituto de Bioquímica y Microbiología, Facultad de Ciencias, Universidad Austral de Chile , Valdivia , Chile
Souza Valeria
Electronic publication date: 2022 Oct 5
Publication date: 2022
Volume: 10
Electronic Location ID: e14047
Received 2021 Sep 2; Accepted 2022 Aug 22
Copyright: © 2022 Schön et al.
Copyright year: 2022
Copyright holder: Schön et al.
License: This is an open access article distributed under the terms of the Creative Commons Attribution License, which permits unrestricted use, distribution, reproduction and adaptation in any medium and for any purpose provided that it is properly attributed. For attribution, the original author(s), title, publication source (PeerJ) and either DOI or URL of the article must be cited.
License URL: https://creativecommons.org/licenses/by/4.0/

Keywords: Bistorta vivipara, Belowground fungal communities, Sebacinales, Diversity, Species hypotheses, Generalist, Alpine knotweed, ITS, Sanger sequencing, Global scale

Funding: German Research Foundation (DFG) 150877455 Estonian Research Council Grant PRG1170 This research was financially supported by the German Research Foundation (DFG), Grant 150877455 as well as by the Max Planck Society. The work of Kessy Abarenkov was supported by the Estonian Research Council grant (PRG1170). The funders had no role in study design, data collection and analysis, decision to publish, or preparation of the manuscript.

==============================
Bistorta vivipara is a widespread herbaceous perennial plant with a discontinuous pattern of distribution in arctic, alpine, subalpine and boreal habitats across the northern Hemisphere. Studies of the fungi associated with the roots of B. vivipara have mainly been conducted in arctic and alpine ecosystems. This study examined the fungal diversity and specificity from root tips of B. vivipara in two local mountain ecosystems as well as on a global scale. Sequences were generated by Sanger sequencing of the internal transcribed spacer (ITS) region followed by an analysis of accurately annotated nuclear segments including ITS1-5.8S-ITS2 sequences available from public databases. In total, 181 different UNITE species hypotheses (SHs) were detected to be fungi associated with B. vivipara, 73 of which occurred in the Bavarian Alps and nine in the Swabian Alps–with one SH shared among both mountains. In both sites as well as in additional public data, individuals of B. vivipara were found to contain phylogenetically diverse fungi, with the Basidiomycota, represented by the Thelephorales and Sebacinales, being the most dominant. A comparative analysis of the diversity of the Sebacinales associated with B. vivipara and other co-occurring plant genera showed that the highest number of sebacinoid SHs were associated with Quercus and Pinus, followed by Bistorta. A comparison of B. vivipara with plant families such as Ericaceae, Fagaceae, Orchidaceae, and Pinaceae showed a clear trend: Only a few species were specific to B. vivipara and a large number of SHs were shared with other co-occurring non-B. vivipara plant species. In Sebacinales, the majority of SHs associated with B. vivipara belonged to the ectomycorrhiza (ECM)-forming Sebacinaceae, with fewer SHs belonging to the Serendipitaceae encompassing diverse ericoid–orchid–ECM–endophytic associations. The large proportion of non-host-specific fungi able to form a symbiosis with other non-B. vivipara plants could suggest that the high fungal diversity in B. vivipara comes from an active recruitment of their associates from the co-occurring vegetation. The non-host-specificity suggests that this strategy may offer ecological advantages; specifically, linkages with generalist rather than specialist fungi. Proximity to co-occurring non-B. vivipara plants can maximise the fitness of B. vivipara, allowing more rapid and easy colonisation of the available habitats.

Introduction

Fungi play a critical role in a wide range of ecosystems either as decomposers allowing nutrient recycling, parasites or commensals, or by establishing mutualist interactions with plants. Within the mutualist interactions, a highly diverse and widely distributed group of fungi form symbioses known as ectomycorrhizae (ECM) with the roots of various plant species in boreal, temperate and arctic regions. Investigations exploring and investigating the mycorrhizal status in plants and its occurrence in terrestrial ecosystems have revealed that ECMs are common in some herbaceous plant genera belonging to the Cyperaceae, Rosaceae, and Polygonaceae (Haselwandter & Read, 1980; Massicotte et al., 1998; Schadt & Schmidt, 2001; Mühlmann & Peintner, 2008a; Gao & Yang, 2016). Out of these, Bistorta vivipara (L.) Delarbre (syn.: Polygonum viviparum; Polygonaceae) with a disjoint arctic and alpine distribution across Europe and Northern America, and temperate and tropical Asia (Marr et al., 2013) has the ability to establish an ECM interaction with a wide range of fungi, enabling it to function as a pioneer plant in stressful habitats (e.g., Brevik et al., 2010). Several studies, including those of Harley & Harley (1987), Massicotte et al. (1998), Cripps & Eddington (2005), Mühlmann, Bacher & Peintner (2008) and Kauserud et al. (2012) have reviewed the earlier literature on morphological studies of these symbiotic interactions.

In the last 20 years, molecular approaches have detected that the belowground diversity of fungi associated with B. vivipara is much greater than that observed only by sporocarps. Using Sanger sequencing, Mühlmann, Bacher & Peintner (2008) detected a high heterogeneity in the ECM community of an alpine primary successional glacier forefront, and Brevik et al. (2010) found that the ECM diversity was higher in stabilised vegetation than in pioneer vegetation, whereas Thoen et al. (2019) detected spatial structuring of the root-associated fungi within the root system. On the other hand, studies using next-generation sequencing have found high patchiness in fungal communities along a primary succession gradient (Blaalid et al., 2012). Other observations have shown that a turnover in fungal community composition along an alpine ridge-to-snow gradient was not linked with increasing or decreasing EMC species richness (Yao et al., 2013), and that there was a decrease in per-plot plant molecular operational taxonomic unit richness with increasing latitude and a non-spatial autocorrelation between sites (Blaalid et al., 2014). Botnen et al. (2019) found that B. vivipara root-associated fungal communities exhibited strong biogeographical structuring, and both compositional variation and fungal species richness correlated with annual temperature and precipitation. Furthermore, the primary succession of B. vivipara root-associated fungi was reported to reflect that of the co-occurring vegetation (Davey et al., 2015). At a fine scale, the root-associated communities are driven by factors of the neighbouring ECM plants (Mundra et al., 2015b) and differences between summer and winter months (Mundra et al., 2015a). Botnen et al. (2014) revealed that B. vivipara and other widely distributed arctic plant species share several similar root-associated fungi.

Fungi belonging to the phyla Glomeromycota (Eriksen, Bjurek & Dhillio, 2002), Ascomycota (e.g., Pezizales and Cenococcum geophilum), and Basidiomycota (Agaricales, Sebacinales and Thelephorales) have been reported to be commonly associated with B. vivipara (see e.g., Mühlmann, Bacher & Peintner, 2008; Davey et al., 2015; Mundra et al., 2015a, 2015b). Because members of the Sebacinales are widely distributed and common in a variety of habitats with different functional roles role ranging from ECM and orchid mycorrhiza to endophytic interactions with leafy liverworts and a wide group of other plants, they are an attractive group to explore fungal diversity, composition, and host-plant specificity at different spatial scales (Garnica et al., 2013; Riess et al., 2014; Mundra et al., 2015a; Garnica et al., 2016a; Arraiano-Castilho et al., 2021). We have extensive knowledge of the occurrence of Sebacinales either from fruiting bodies or from fungal sequencing from roots of various plant groups in the Bavarian Alps (Garnica et al., 2013; Riess et al., 2014; Arraiano-Castilho et al., 2021). However, despite these important contributions, there are still significant unanswered questions concerning the fungal communities associated with the roots of B. vivipara. Since this plant species has a moderately long lifespan—according to Callaghan & Collins (1981), individuals of B. vivipara can reach an age of 27+ years—one important question is if the co-occurring vegetation (especially trees that can live some hundreds of years) may serve as a reservoir of ECM fungi. According to this perspective and based on the active symbiont recruitment from neighbouring plants reported by Thoen et al. (2019), we hypothesise that B. vivipara should have the same fungi as the plants present in its surroundings and we therefore would expect a similar associated ECM diversity and composition. To test this hypothesis, we generated new Sanger sequences data for fungi associated with the roots of populations of B. vivipara from a site in the Bavarian Alps and also a site in the Swabian Alps following the natural range of distribution of this plant species in Germany (Fig. 1A) and analysed them together with sequences from public databases. We first determined and compared the ECM diversity associated with roots of B. vivipara at local, regional, and global scales. We then compared the diversity and composition of root-associated fungi between B. vivipara and other plant species in a global dataset of Sebacinales. In addition, we examined how much of the Sebacinales diversity associated with B. vivipara is shared with other host plant species in a global dataset encompassing many co-occurring plant families.

Figure 1 Natural occurrence of Bistorta vivipara in southern Germany.

(A) Map showing the natural occurrence sites in Germany of the populations of B. vivipara. Image credit: Phytodiversität Deutschland & Bundesamt für Naturschutz (Hrsg.) 2013. Verbreitungsatlas der Farn- und Blütenpflanzen Deutschlands, Landwirtschaftsverlag, Münster (ISBN 978-3-7843-5319-7). Records before 1950 (red) and after 1980 (black). (B) Collection site (timberline) in the Bavarian Alps (Mt. Isler) (right arrow). (C) Collection site in the Swabian Alps (left arrow). (D) Individual B. vivipara specimen with flowers.

Materials and Methods

Collection sites

Root-associated fungi were surveyed from B. vivipara plants collected in the Bavarian Alps along a short (500 m) altitudinal gradient on the northern and southern slopes of Mt. Iseler (1,876 m) in the region of Bad Hindelang-Oberjoch in Germany (47°29′57″N, 10°25′5′E) (Fig. 1B). The soil layer consists predominantly of intense lithoidal calcareous brown soils (secondary podzols and chromic luvisols) with an average thickness of >60 cm (Bräker, 2000), whereas the deeper soil layers are mainly characterised by basic dolomite, a sedimentary carbonate rock consisting of the mineral calcium magnesium carbonate (Freudenberger & Schwerd, 1996). Mt. Iseler’s northern and southern slopes are subject to seasonal pasture grazing by cattle. At disturbed sites, B. vivipara grows especially well in cattle hoof prints and narrow trails. The total annual rainfall is 1,799 mm (1961–1990), with maximum levels from May to August (228.1 mm/month) and minimum levels from September to October (98.9 mm/month). The annual average air temperature is 5.9 °C at an altitude of 1,136 m, with a maximum of 14.4 °C during July and a minimum of –1.8 °C in January. The average annual snow cover is around 150 days from October to mid-April.

Furthermore, individuals of B. vivipara were also collected from the Swabian Alps located in the Irndorfer Hardt Nature Reserve (Baden-Württemberg) near Tuttlingen, Germany (8°94′50.14″N, 48°09′00.28″E) (Fig. 1C). The nature reserve is located between 855 and 880 m above sea level and has a surface area of 103 ha (Döler & Detzel, 2008). Although calcareous soils are found on the hills and on the slopes of Irndorfer Hardt, the shallow portion of the valley is covered with deep and decalcified clay soil. The annual average temperature is 6 °C, with extreme temperatures reaching –30 °C in winter (Wilmanns, 2005). In general, the reserve is characterised by having a long period of snow cover in winter. Hoarfrost or frost is likely to occur throughout the year. These low temperatures are the result of a low-pressure trough that hinders the flow of cool air away from the area. The Authority for Unit Environment from the Government of Freiburg (Germany) granted permission to conduct sampling at the Natural Reserve Irndorfer Hardt.

Sample collection and processing of root samples

Samples were collected from Mt. Iseler in the Bavarian Alps at the end of July 2012. At the two lowest collection sites, B. vivipara occurred in disturbed meadows (which are used by cattle farmers), whereas at the highest sites it occurred in pristine mountain habitats. Along the altitudinal transect, we randomly sampled 80 individuals of B. vivipara at different stages of development (rooted bulbils to fully flowered plants) from 16 patches.

In the Swabian Alps, we were, with permission from the local nature reserve authorities, only allowed to collect 10 individuals of B. vivipara (August 2011). The reserve harbours a small relict population under strong protection. Plants were sampled from three sites at a distance of approx. 300 m to 800 m from each other.

Plants with intact root systems and a portion of soil were carefully excavated, placed in plastic bags, and stored in a cooler for transport to the laboratory. In the laboratory, plants were placed in buckets containing tap water for a few hours before roots were rinsed with additional tap water to remove soil particles and foreign roots. Subsequently, B. vivipara roots were rinsed with sterile ddH2O at least five times. From each plant collected in Bavarian Alps, ten ECM root tips were randomly selected and cut from each plant with a sterile razor blade and separated from the others with forceps. As some plants from the Swabian Alps had a small root system, a smaller number of root tips per individual were taken. ECM root tips were then classified primarily on the basis of their morphology, i.e., ramification type, size, colour, presence of emanating hyphae and rhizomorphs (Agerer, 1987–1997). To analyse the widest range of ECM diversity, we selected morphotypes covering the complete spectrum of forms and colours. In addition, for those dominant morphotypes were arbitrarily chosen two to five ECM root tips from each morphotype. Individual ECM tips were placed in a sterile 1.5-ml Eppendorf tube, and air dried at 50 °C over night.

DNA extraction, PCR, cloning, DNA sequencing and sequence editing

In total, we processed 800 root tips of B. vivipara (5 plant individuals × 16 patches × 10 morphotypes) from the Bavarian Alps and a total of 57 root tips from the Swabian Alps (10 plant individuals × 2 patches, as some plants had a root system that was too small, we were unable to collect 10 root tips). Thus, genomic DNA was extracted from 857 ECM root tips with the innuPREP Plant DNA Kit (Analytik Jena AG, Jena, Germany), according to the manufacturer’s instructions. Dried samples were placed in liquid nitrogen and ground to a fine powder with sterile pestles. PCR reactions and thermal profiles were performed as described in Garnica et al. (2016b). Negative controls lacking DNA were used in all PCRs. For the detection of B. vivipara mycobionts, we amplified the internal transcribed spacers (ITS1 and ITS2) and 5.8S rDNA with the primers ITS1F (Gardes & Bruns, 1993) and ITS4 (White et al., 1990).

PCR products were cleaned with a diluted 1:20 ExoSAP-IT purification kit (USB Corporation, Cleveland, OH, USA), following the manufacturer’s instructions. In the case of multiple ITS amplicons or where the PCR products could not be sequenced directly, they were cloned according to Garnica et al. (2016a). DNA amplifications of selected colonies were used directly as templates for PCR with MangoTaq DNA-polymerase (Bioline, London, England), and the M13 forward and reverse primers (Invitrogen, Waltham, MA, USA). From each DNA extraction, eight bacterial colonies were used for PCR reactions. Amplicons containing cloning inserts were diluted 1:20 before purification.

Both DNA strands were cycle-sequenced with the same PCR primer combination used in the PCR amplifications. In cases where the sequence quality was too low, we additionally sequenced the PCR products with the highly universal DNA barcoding primers ITS2 (White et al., 1990) and 5.8SR (Vilgalys & Hester, 1990) and ITS3Seb for Sebacinales (Berbee, in Setaro et al., 2006). Cycle sequencing was carried out with 1 µL of the sequencing primer in combination with 4 µL of a dye terminator sequencing kit (BigDye 3.1; Applied Biosystems, Foster City, CA, USA) diluted 1:6 and 5 µL of the purified DNA template. Cycling products were precipitated with isopropanol (75%), purified with ethanol (80%), and dried in a vacuum centrifuge.

DNA strands were sequenced on an ABI Prism 3130xl automated sequencer (Applied Biosystems, Foster City, CA, USA). Forward and reverse sequence chromatograms were assembled and manually edited with Sequencher (Gene Codes Corporation, Ann Arbor, MI, USA). Plant specimens yielding DNA of ECM fungi were deposited in the Herbarium Tubingense.

All sequences forming ECM generated in this study are available in GenBank under accession numbers KF000466–KF000687 and KC986236–KC986286.

Fungal diversity and composition associated with Bistorta vivipara

Well-annotated public fungal rDNA ITS sequences associated with B. vivipara as a host were downloaded via the PlutoF platform (https://plutof.ut.ee) on 05-02-2022. All sequences available from the International Nucleotide Sequence Databases (http://www.insdc.org/) and the UNITE database were included. These data come from different locations, but are restricted to European sites, mainly from Norway, Germany and Estonia (Table S1). We performed quality control and data cleaning steps as well as the extraction of the full ITS1-5.8S-ITS2 region with ITSx (Bengtsson-Palme et al., 2013) for further data analysis. The sequences were then subjected to hierarchical clustering with the 1.0% distance threshold as used by UNITE (Kõljalg et al., 2013) and implemented in the species hypothesis (SH) matching analysis tool available on the PlutoF workbench. This dataset also includes all full-length ITS sequences obtained in this study.

Using this dataset, we counted the number of SHs that were observed in either one of the two sites sampled here (Swabian Alps, Bavarian Alps) or in any other location, as well as the overlaps between these three groups. Furthermore, we compared the relative abundance of fungal orders for these three groups in order to identify common dominant fungal partners of B. vivipara over all of its sampled natural range.

Sebacinales diversity associated with Bistorta vivipara and other plants

Similarly to the above described workflow, we downloaded well-annotated public rDNA ITS sequences of Sebacinales via the PlutoF platform (https://plutof.ut.ee) on 29-09-2020. These included sequences from the studies by Brevik et al. (2010) from Norway (Svalbard archipelago), by Mühlmann & Peintner (2008b) from Austria (Tyrolean Alps) and Garnica et al. (2013) from Germany (Bavarian Alps), and those from other papers, available from the International Nucleotide Sequence Databases (http://www.insdc.org/) and the UNITE database (Kõljalg et al., 2005; https://unite.ut.ee/). The PlutoF platform (Abarenkov et al., 2010) provides third-party annotation possibilities for DNA sequences available in International Nucleotide Sequence Databases and UNITE (Nilsson et al., 2019) and includes metadata on the country of origin and interacting taxa, which are often missing in the original data sources. Quality control, ITS extraction and hierarchical clustering were performed as described above.

We then focused on the Sebacinales fungi associated with B. vivipara and other plant hosts. For this, we used the data generated here, our own data from previous studies, and the public data described above. We counted the number of SHs and sequences per genus of plant host that co-occurs with B. vivipara. To show any potentially co-occurring plants, we were permissive in this definition, including all plant genera that occur in the northern hemisphere. We further normalised the number of SHs with the number of sequences per plant genus in order to reduce the bias of uneven sampling across plant genera.

For the interactive Tree Of Life (iTOL) ITS tree (Kozlov et al., 2019; Letunic & Bork, 2021), all Sebacinales sequences were subjected to three-step clustering (90%, 95%, and 97% sequence similarity) via vsearch v2.15.0 (Rognes et al., 2016). In all clustering steps, for clusters that contained at least one sequence from B. vivipara, all sequences were kept, whereas for clusters without any sequence from B. vivipara, only one representative per cluster was retained. Because of the high number of originating countries and the great diversity of host taxa, we decided to colour-code the continent of the country of origin and the plant host class, marking sequences from B. vivipara with a distinct colour.

Finally, we counted the amount of SHs associated with B. vivipara that were shared between two plant host families and visualised these numbers as a circular network, with the width of the connecting edges representing the number of pairwise shared SHs.

Results

Diversity and composition of fungi associated with the roots of B. vivipara in the Bavarian Alps, the Swabian Alps, and other European sites

In total, 332 non-chimerical fungal ITS sequences, clustered into 90 different SHs, were detected as associated with B. vivipara from the two study sites in the Bavarian Alps and the Swabian Alps (Table S1).

From Mt. Iseler in the Bavarian Alps, 264 ITS sequences representing 73 SHs were obtained from B. vivipara roots. Members of the Basidiomycota were represented as follows: Thelephorales, 30 SHs (94 sequences); Sebacinales, 24 SHs (114 sequences); Agaricales, six SHs (20 sequences). In the Ascomycota, the Pezizales were represented by nine SHs (30 sequences); Helotiales by three SHs (four sequences), and Mytilinidales (Cenococcum) by a single SH (two sequences) (Fig. 2A).

Figure 2 Fungal diversity and composition associated with Bistorta vivipara.

(A) Relative abundance of fungal orders in the Bavarian Alps, the Swabian Alps and other sites globally. Only the orders that are present at the sites studied here are shown for clarity. (B) Number of fungal species hypotheses (SHs) associated with B. vivipara in two mountain sites located in the Bavarian Alps and the Swabian Alps (Germany), as well as in other sites globally (combined). The number of SHs that are shared between these three groups (Bavarian Alps-Other and Bavarian Alps-Swabian Alps) is also shown.

In the Swabian Alps, 68 ITS sequences were obtained from B. vivipara roots which clustered into nine SHs. The Basidiomycota were represented as follows: Sebacinales, three SHs (11 sequences); Thelephorales, three SHs (44 sequences); Russulales, one SH (two sequences), and Agaricales, one SH (one sequence). The Ascomycota were represented by Pezizales with one SH (10 sequences) (Fig. 2A).

From other publicly available data sources, covering a wide range of European distribution of B. vivipara, we identified 123 fungal ITS sequences distributed over 111 SHs. The most frequently identified fungal orders were, in the Basidiomycota, the Thelephorales (26 SHs, 30 sequences), the Sebacinales (15 SHs, 16 sequences), and the Agaricales (15 SHs, 15 sequences). Ascomycota were represented by Helotiales (14 SHs, 15 sequences) and Pezizales (8 SHs, nine sequences).

In terms of overlaps between sites, most SHs were only detected in a single location, but one SH was shared between the Bavarian Alps and the Swabian Alps (Sebacinales SH 1800251). A total of 11 SHs were shared between the Bavarian Alps and other sites (six Thelephorales SHs, three Sebacinales SHs, one Helotiales, and one Mytilinidales SH) (Fig. 2B).

Diversity of Sebacinales associated with Bistorta and other plants

In total, 5,650 Sebacinales ITS sequences clustered in 1,520 SHs with a cutoff of 99% (Table S2). For 4,500 ITS sequences, a match to 1,350 known SHs from the UNITE database was found, whereas 1,150 ITS sequences had no match to a known SH (170 SHs in total). Of the ITS sequences generated in this study, 36 ITS sequences had no known SHs. The Sebacinales sequences were associated with the roots of 240 different plant genera (Fig. 3 and Table S3). The plant genera with the highest number of associated fungal SHs were Quercus (149 SHs with 214 sequences), Pinus (102 SHs with 139 sequences), and Bistorta (54 SHs, whereof 49 with B. vivipara and 5 with B. officinalis, with 176 sequences).

Figure 3 Sebacinales diversity associated with B. vivipara and other plants co-occurring with B. vivipara.

(A) Number of species hypotheses (SHs) per plant genus. (B) Number of Sebacinales sequences per plant genus. (C) Number of SHs vs number of Sebacinales sequences (seqs) per plant genus.

A very diverse group of Sebacinales SHs from the Sebacinaceae family with 467 SHs was found to be associated as ECM with B. vivipara; a smaller number of SHs came from the Serendipitaceae family involved in cavendishioid, ericoid, orchid, and ECM interactions (52 SHs, see Fig. S1 and Table S4).

Co-occurrence of sebacinoid SHs with B. vivipara and other plant hosts

Of the 49 sebacinoid SHs associated with B. vivipara, with members of Fagaceae were shared 22 SHs, with Pinaceae 16 SHs, with Rosaceae 15 SHs, with Salicaceae 12 SHs, with Betulaceae 11 SHs, and with Orchidaceae 10 SHs (Fig. 4). One SH (SH1723664) was associated with 14 different plant host families, followed by two SHs (SH1731002 and SH1731004) that were associated with 12 and 11 plant hosts, respectively.

Figure 4 Number of Sebacinales SHs shared between Bistorta vivipara and other host plant families.

The width of each edge is proportional to the number of SHs shared between the two plant family nodes it connects. The size of the nodes is proportional to the number of SHs associated with B. vivipara that has been observed with that plant family.

Discussion

Harsh environments promote generalist fungal communities more adapted to these conditions rather than plant-specific specialists (Kernaghan & Harper, 2001; Ryberg, Andreasen & Björk, 2011; Botnen et al., 2014; Brunner et al., 2017). Several common fungal lineages in these environments are widely distributed and can occur in several plant communities (Ryberg, Andreasen & Björk, 2011). Thus, the high proportion of non-host-specific fungi is advantageous for B. vivipara plants specifically because they can recruit all the available fungi and thus ensure their survival under unfavourable conditions (Kernaghan & Harper, 2001). In this context, because most studies have focused on the diversity of fungal communities associated with B. vivipara from artic, subarctic, and mountainous regions, we expand on this issue here by studying two mountain ecosystems as well as a more global distribution by comparing B. vivipara with other plant hosts and focus on a phylogenetically diverse and widely distributed fungal lineage, the Sebacinales.

Fungal communities associated with B. vivipara are diverse but stable over large geographic distances

Both mountainous study sites, the Bavarian Alps and the Swabian Alps, harbour populations of B. vivipara that are characterised by a phylogenetically diverse array of symbionts, mainly from the group of ECM-forming fungi, in line with previous morphological studies (e.g., Harley & Harley, 1987; Massicotte et al., 1998). Our results confirm the previously suggested dominance of Agaricales, Sebacinales and Thelephorales (all Basidiomycota), Pezizales and Cenococcum geophilum (Ascomycota) in the ECM communities of various plant species (Mühlmann & Peintner, 2008b; Mühlmann, Bacher & Peintner, 2008; Bjorbækmo et al., 2010; Peintner & Kuhnert, 2010; Blaalid et al., 2012). Representatives of the Thelephorales were the most frequent group of ECM fungi associated with B. vivipara. Their relevance and importance as ECM-forming fungi was long underestimated (Koljalg et al., 2000). Sebacinoid fungi formed the second largest group of ECM fungi, both in terms of the number of SHs and of sequences. Previous research reported an abundance of Pezizales ECM with B. vivipara (Tedersoo et al., 2003; Izzo, Agbowo & Bruns, 2005), which was also confirmed here. Most of the identified Pezizales sequences matched sequences from sporocarp tissue of the genera Genea, Peziza, Sarcosphaera, Tarzetta, and Trichophaea, which are already known as major representatives of the Pezizales from northern temperate forests (Tedersoo et al., 2006). In spite of the high ECM diversity detected, the non-asymptotic shape of the species accumulation curve suggests that many fungal species remained undetected (Fig. S2).

When comparing the dominant fungal groups of the two studied sites with data from B. vivipara available from public data sources, a strikingly similar pattern is observable. In this dataset, Thelephorales, Sebacinales, and Agaricales are also the most abundant representatives of the Basidiomycota, while Ascomycota are most often represented by Pezizales or Helotiales. While these fungal groups generally dominate ECM communities of many plants, this could also indicate active selection by B. vivipara. However, since the public data only covers (Central and Northern) Europe, we cannot make any inferences about the diversity and composition of fungal communities associated with B. vivipara across its whole natural distribution.

It has been hypothesised that B. vivipara originated in Asia, followed by relatively recent expansions to North America and Europe before the Last Glacial Maximum (Marr et al., 2013). Prior to and during the Last Glacial Maximum, habitat modification affected the dispersal patterns of many plant species (Hewitt, 2000; Birks, 2008). In Europe, macrofossils (bulbils) of B. vivipara in sediments collected in Southern Norway were dated to around 11,000 years ago (Birks & Van Dinter, 2010). In this study, our two study sites (Swabian Alps and the Bavarian Alps) spanned the natural distribution range of two haplotypes of B. vivipara (Marr et al., 2013). Here we found evidence of several shared SHs among the two studied populations of B. vivipara and between the Bavarian alps and other European locations. Furthermore, the composition of associated fungi in terms of the major groups closely resembled each other in all three groups. This could indicate the co-migration of host plants and some of their associated fungi throughout Europe. Moreover, it seems possible that more intense sampling and advanced sequencing techniques could reveal additional SHs shared between these regions.

Sebacinales are a diverse group associated with B. vivipara and other plants

Our analyses reveal that Sebacinales form a major part of the fungal diversity associated with B. vivipara, and that the diversity of Sebacinales associated with it is comparable to that observed for several tree species. While most of the sebacinoid associations seem to be ECM, other types of interactions between Sebacinales and B. vivipara have also been reported. Previously, Riess et al. (2014) and Garnica et al. (2013) detected the presence of Sebacinales belonging to the Serendipitaceae associated with B. vivipara. In this study, the presence of Sebacinales specifically belonging to the family Serendipitaceae was also detected for B. vivipara (Fig. S1). This family encompasses endophytic, ericoid, orchid, and ectomycorrhizal fungi (Oberwinkler et al., 2013). Future studies integrating data from both full-length ITS and ITS1/ITS2 (i.e., also taking into account short-read data from e.g., Blaalid et al., 2012, 2014; Botnen et al., 2014; Davey et al., 2015; Mundra et al., 2015a, 2015b; Arraiano-Castilho et al., 2021) are likely to give more insights into the different types of associations between Sebacinales and B. vivipara.

Sebacinoid fungi associated with B. vivipara are likely recruited from co-occurring plants

Similar to observations in other plants (Bahram et al., 2012; DeBellis et al., 2006; Bogar & Kennedy, 2013), our study indicates that the diversity of sebacinoid fungi is greatly influenced by the presence of co-occurring plant species, mainly ECM trees, suggesting that unspecific mycobionts are able to colonise B. vivipara when it grows near other ECM-forming plants. In general, an enormous diversity of symbionts is involved in ECM associations with B. vivipara, as supported by the large number of SHs from the Sebacinaceae and Serendipitaceae revealed in this study. The unspecific nature of many of these interactions, combined with stochastic processes of colonisation mediated by the bulb dispersion, for example through birds (Moss & Parkinson, 1975), and/or the patchiness of fungal distributions, might be responsible for the very heterogeneous ECM communities detected in this study.

In accordance with a previous study by Botnen et al. (2014) that revealed the low host specificity of ECM communities associated with B. vivipara, we detected many SHs that were shared between B. vivipara and both ECM-forming and non-ECM plants in the Sebacinales. In fact, only 4 non-singleton SHs were found to be exclusive to B. vivipara. However, while SHs found with B. vivipara are mostly shared with other plants, our data suggests that ECM-forming trees associate mostly with fungal partners not shared with other plants. This could be ascribed to a higher fraction of specific fungal partners of these plants, or to sampling bias and/or the molecular approach applied. For example, considering the enormous root systems of P. abies, it is difficult to detect the complete ECM diversity from a few samples. In B. vivipara, in contrast, its small size makes it easier to investigate whole root systems with relatively little sampling effort. Our results also confirm those of Riess et al. (2013), indicating high cryptic diversity in Sebacinales.

Conclusions

Our findings at the local and continental levels reinforce previous observations in B. vivipara and other plant groups, that the presence of surrounding vegetation may serve as reservoirs of fungal symbionts. Members of the order Sebacinales represent a frequent and dominant group of fungi associated with B. vivipara. Most Sebacinales species form ECM associations, but some, specifically those from the Serendipitaceae family, can also form endophytic and different mycorrhizal interactions. Few sebacinoids are associated specifically with B. vivipara, whereas most are shared with co-occurring plants. Here, we also present the first fungal ITS sequences from B. vivipara from the Swabian Alps. The data from this site, though obtained by limited sampling, reveals similar taxonomic groups of fungi associated with B. vivipara and reflects the high amount of shared fungal partners observed in more densely sampled sites. However, a more complete sampling of B. vivipara across its natural distribution range, with a locally finer scale and more intense sampling, and the application of high-throughput studies such as metabarcoding could further elucidate major patterns of fungal diversity and composition at both the local and global scale.

Supplemental Information

Supplemental Information 1 iTOL ITS tree with representatives from the full Sebacinales rDNA ITS sequence dataset.

Sequences from the full dataset were subjected to hierarchical clustering to denser sampling of the dataset near the B. vivipara-related sequences and more sparse sampling elsewhere on the tree (see Supplementary Item 2 for a more detailed description of the clustering process). Because of the high number of originating countries and the great diversity of host taxa, the continents of the country of origin and the plant host classes were colour-coded and sequences from B. vivipara were marked with a distinct colour.

Click here for additional data file.

Supplemental Information 2 Species accumulation curve for ECM fungal communities associated with Bistorta vivipara.

Click here for additional data file.

Supplemental Information 3 Species hypothesis (SH) matching analysis results for the dataset of 455 quality-filtered ITS sequences associated with B. vivipara.

Assignment of sequences to fungal species hypotheses using hierarchical clustering on a 1.0% distance threshold.

Click here for additional data file.

Supplemental Information 4 Species hypothesis (SH) matching analysis results for the dataset of 5650 quality-filtered Sebacinales ITS sequences used in the current study.

Assignment of sequences to fungal species hypotheses using hierarchical clustering on a 1.0% distance threshold.

Click here for additional data file.

Supplemental Information 5 Number of sebacinoid species hypotheses (SHs) and sequences associated with B. vivipara and other plant genera.

Click here for additional data file.

Supplemental Information 6 Species hypothesis (SH) matching analysis results for the dataset of 1155 quality-filtered Sebacinales ITS sequences used in the iTOL phylogenetic tree.

Click here for additional data file.

Supplemental Information 7 RAxML (Kozlov et al., 2019) phylogenetic tree prepared with the iTOL tool (Letunic & Bork, 2019; https://itol.embl.de/) with representatives from the full Sebacinales rDNA ITS sequence dataset.

All sequences were subjected to three-step clustering (90%, 95%, and 97% sequence similarity) via vsearch v2.15.0 (Rognes et al., 2016). In all the clustering steps, for clusters that contained at least one sequence from B. vivipara, all sequences were kept, whereas for clusters without any sequence from B. vivipara, only one representative per cluster was retained at the 95% and 97% sequence similarity levels. Because of the high number of originating countries and the great diversity of host taxa, we decided to colour-code the continent of the country of origin (green, Europe; blue, North America; pink, Asia; grey, South America; orange, Pacific; yellow, Africa; red, unspecified) and host class while marking sequences from B. vivipara with a distinct colour. Sequence labels on the tree consist of three distinct parts with only the middle part being mandatory: (1) sequence taxon name, (2) sequence accession number and the UNITE species hypothesis (SH) code it belongs to (either existing or new; the latter are marked with # or #s and a number concatenated with a slash) and (3) sequence count for the number of sequences behind this representative. Each representative represents sequences from the same country and host family belonging to the same SH at a 1.0% distance threshold.

Click here for additional data file.

We thank S. Silberhorn for assistance with the laboratory work. Sequence data from the Swabian Alps was generated by J. Schade as part of her Bachelor thesis, data from the Bavarian Alps was generated by P. Nassal as part of his Diploma thesis. We also thank K. Riess for help with the collection of material and by providing a picture of B. vivipara, and R. May for providing the map with the natural distribution of Bistorta vivipara in Germany. We thank the two anonymous reviewers whose comments/suggestions helped improve and clarify this manuscript.

Additional Information and Declarations

Competing Interests

Author Contributions

Field Study Permissions

DNA Deposition

Data Availability

The authors declare that they have no competing interests.

Max Emil Schön conceived and designed the experiments, performed the experiments, analyzed the data, prepared figures and/or tables, authored or reviewed drafts of the article, and approved the final draft.

Kessy Abarenkov performed the experiments, analyzed the data, prepared figures and/or tables, authored or reviewed drafts of the article, and approved the final draft.

Sigisfredo Garnica conceived and designed the experiments, performed the experiments, prepared figures and/or tables, authored or reviewed drafts of the article, and approved the final draft.

The following information was supplied relating to field study approvals (i.e., approving body and any reference numbers):

Permission from the government of Freiburg, Unit Environment (Germany) was received for sampling in the Natural Reserve Irndorfer Hardt.

The following information was supplied regarding the deposition of DNA sequences:

The data is available at GenBank: KF000466–KF000687 and KC986236–KC986286.

The following information was supplied regarding data availability:

The raw data from the phylogenetic tree are available in the Supplemental File. Custom scripts are available at https://github.com/maxemil/bistorta-vivipara.

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
