# Peer review of "Host generalists dominate fungal communities associated with alpine knotweed roots: a study of Sebacinales"

_PeerJ, doi:10.7717/peerj.14047_

## Round 0.1 · original submission · Major Revisions

Both reviewers had constructive criticism about the study design and pointed out some ambiguity of the motivation for the study and specific questions being addressed. One reviewer pointed out the omission of a relevant recent publication and dataset that needs to be included for comparison. A number of other, more minor, issues were pointed out by both reviewers. Taken together, while the study certainly has merit, both reviewers felt that more effort was needed to better clarify the questions being addressed and some weaknesses of the study design that compromise the interpretation of results need to be addressed.

Reviewer 1 ·

Basic reporting

The English used is clear and mostly unambiguous
The references are up to date.
However, there is not enough background provided to understand the motivation for the data analysis.
The article structure is o.k. but important results are not presented, and additional data should be provided in the Supplementary information.

Experimental design

The aims are not clearly formulated
Two sites are compared with a completely different sampling depth.
There is no real altitudinal gradient sampled, at least it is not well-presented as such.
The data analysis is not based on a solid hypothesis.

Validity of the findings

Issues treated in the paper

o The MS investigates the ectomycorrhizal communities associated to Bistorta vivipara in Germany. One sites was extensively sampled once with 80 plants, the other was used as comparison (n=10 plants).
o Based on datamining, ECM communities associated to B. vivipara are compared to ECM communities of other host plants.
o B. vivipara ECM communities are more extensively discussed concerning their second most abundand group – the Sebacinales.


Importance
The work brings B. vivivara ECM in a wider context by comparing local data from Germany with data from surrounding habitats in Europe. It provides a general view of B. vivivara ECM communities and allows for biogeographic comparison and evolutionary insights.

Strength
Dataset of all Bistorta ECM studies is a solid base for comparison of the newly ECM generated data with data from other habitats, and with other, co-occurring ECM.
Dataset on Sebacinales allows – in addition - for a more functional interpretation of Bistorta ECM communities.

Weakness
The question are not well specified, the date analysis has a huge potential, but stays at the surface. Results are often not clearly provided, and the discussion is sometimes speculative:

The paper is not really focussed and presents several issues without deepth. This is a pity, as there is lot of potantial.
Great dataset, but not really well used. The question are not well formulated nor really answered, leaving the reader unsatisfied as no clear statement is made nor new conclusions are drawn.

• How is Bistorta ECR diversity increasing from local-regional-global scale?
• How are ECM communities changing over these scales in composition?
• What is the involvement of Sebacinales in these issues?


Figures

Figure 1
Move distribution map for supplementary information.
Figure 3
This is a large figure with too many plants included. What are the criteria for the plants selected?
Please show only plants which are co-occurring with Bistorta vivipara in the Alpen/montane sites of Europe.
Figure 5:
cannot be interpreted, too small and full with colurs. Please reduce it in order to clearly and intuitively present an important message / result.


Comments for improvement
→ What is the influence of vivipary on the ECM distribution of this special host? This should be discussed.
→ The main goals are difficult to discriminate as presented here and are not really clear: please formulate less ambiguously
-diversity of fungal communities vs. phylogenetic diversity
→ Please explain more in detail - why the focus on Sebacinales? Why is this important in this context = ECM of B. vivipara.

→ when comparing ECM associated to B. vivipara and other plant species, the number of studies, sampling effort, and geographic distribution should be considered. More sampling sites, wider geographic range will very likely result in higher diversity. This should be considered in the analysis, e.g. by randomization.
Moreover, it is not really clear, why all these plants are compared, as they come from quite different habitats. Is there always an overlap between Bistorta habitats and habitats of these plants? Better explain the hypothesis and results of this analysis.

→ The comparison of the 2 regions, and the hypothesis of co-migration based on 10 sampled plants is highly speculative in this context. This could be done with the dataset, as it includes sequences from disjunct sampling sites of B. vivipara. I would love to see how the associated ECM fungi changed (diversity, composition) and what the different sites in Austria, Scotland, Norway, Germany have in common. This would provide solid data fort he hypothesis and warrant further exploration.
The ECM of B. vivipara in the Swabian Alps are a nice small example, but what is the goal?

→ The data on Sebacinales need to be better presented in the results,! They are barely mentioned. How high is speciess richness of Sebacinales in Bistorta in general?

→ Provide a list of detected SH in the SI

Additional comments

Further comments

line 126 - specify what you intend with long/short vegetation periods

-Materials/methods
Specify how your altitudinal transect was setup-more give detailed information!
-Random selection of plants at diff. stages of development along the transect in Bavarian Api n = 80 in Swabian Alps only n=3×3 (+1?)
-are these date comparable?
-No morphotyping of ECM 's? why? N=857 root tips
- line 251: it is not clear, if all there SHs are also associated to Bistorta. Clarify!
- 258 what do you mean with "other hosts"
-No refence to Fig 5 in results.
Provide better insight in Sebacinales data and results

Line 278 - Specialist are rare in ECM fungi; it is rather the exception and usually a drawback for both, fungal partner and plants. You even show this in Fig 4!

line 282- Why do you consider this ecosystem as extreme?
Discuss this based on the richness u. diversity of B. viripara based on different habitats investigated!

308-ff The comparison of the 2 regions, and the hypothesis of co-migration based on 10 sampled plants is highly speculative in this context. This could be done with the dataset, as it includes sequences from disjunct sampling sites of B. vivipara. I would love to see how the associated ECM fungi changed (diversity, composition) and what the different sites in Austria, Scotland, Norway, Germany have in common. This would provide solid data fort he hypothesis and warrant further exploration.

Consider also patterns of distribution of B. vivipara. Bulbils: birds as important vectors

322-The altitudinal gradient is overstressed.The sampling sites du not really reach high-altitudinal sites. This statement is not supported by the data.

Reviewer 2 ·

Basic reporting

Professional English is used throughout.

Sufficient background is covered in the introduction, although there is no reference to other small ectomycorrhizal plants/families co-occurring with B. vivipara. A relevant study seems to be missing in the reference list (Arraiano-Castilho et al. 2021. New Phytologist 229, 2901–2916). It is not clear if the sequences generated in that study have been included in the analyses in this manuscript. This needs to be clarified and if they haven't, they need to be included and re-analysed with the rest of the sequences, as the study explore and generate sequences of the EM communities in roots of B. vivipara and other two dwarf plants across the same countries included in this study (European Alps).

The structure is correct. Some changes can be made, however, in Figures:
Figure 1. Source can be added as a note below the figure. I cannot see section d) of the figure.
Figure 2. It is hardly representative of the actual and shared diversity, since the sampling effort in terms of individuals sampled for ectomycorrhizas is uneven (80 in the Bavarian Alps and 11 in Swabian Alps). I would move this to supplementary materials.
Figure 3. I think this figure would be more relevant for the content of the manuscript including only co-occurring hosts with B. vivipara,
Figure 4. Have the sequences from the article mentioned above included in here?
Figure 5. As it stands, I am not sure what the authors want to show in this figure. This needs to be explained/highlighted in the figure caption.

Results regarding comparisons or diversity in the two sites samples in Germany should be carefully described and discussed, as the sampling design is uneven. More detailed methods need to be provided (see below).

Experimental design

Authors generate data on fungal communities associated to few individuals of B. vivipara in two mountain locations in Germany (primary research). They use their sequences and add sequences from UNITE/PlutoF to compare host specificity of the fungi that associate with B. vivipara and they also analyse separately sequences belonging to Sebacinales across hosts. The aim for doing this and the process is not very clear to me.

While there are many other studies focusing on the diversity of fungi that associate to B. vivipara, maybe the most interesting part of this article is the comparison of the communities of this plant with other plants (not only small plants above the treeline) but also trees. Although authors mention that they only include co-occuring genera (e.g. L30) I can see e.g. Uapaca and Nothofagus in Fig. 3., can you elaborate/explain on this?

From my point of view, methods need to be more detailed to avoid misinterpretation of the results. For instance, it is not clear how the plant individuals were chosen and how and how many ectomycorrhizas were collected per plant individual; was the sampling effort similar per plant? How did you get to 857 ectomycorrhizas? Please explain.

L176-177: please detailed the primer combination used and for which purpose.
L189-190: when was this done and which version of UNITE was used?
L198-206. This belongs to results
L209-210. It is not clear how the sequences were identified. Were they all assigned one SH number individually? Please explain.
L214-215. Can you please elaborate or reference the source? Did you retrieve all the sequences belonging to Sebacinales from UNITE/PlutoF? Please clarify.
L225. Please spell iTOL.
L233, L249. I think authors mean taxonomic diversity, otherwise, please explain how this is calculated.

Validity of the findings

As mentioned above, the experimental and study design seem to show some caveats that need to be explained/addressed, otherwise some of their conclusions cannot be drawn from the results (e.g. L315-318; L321-323).

Additional comments

L33-34. Which family? Polygonaceae?
Diversity of fungi in B. vivipara roots, at least above the treeline, seems to be mostly driven by soil conditions (e.g. pH and soil nitrogen); pH is one of the main drivers, do you have that information for your sites?
Sections starting in L104 and L129 can be merged.
L200. Is it 1% or 1.5%?. Why do you use a different threshold for Sebacinales?
L198-L206. This should go in results.
L334. There is no overlap with any other plant? Please clarify.

---

## Round 0.2 · Minor Revisions

Please attend carefully to all the reviewers opinions and address the problem with the citation.

Reviewer 2 ·

Basic reporting

Professional English is used throughout the manuscript but the new sections with track changes might need English revision.

I still consider that the authors are omitting a relevant study not only for field background/context but also for their hypotheses and discussion of the results (e.g. (L779-845, L912). Arraiano-Castilho et al. 2021. New Phytologist 229, 2901–2916. Despite using only ITS2 for their analyses, they found 64 OTUs belonging to /sebacina ECM lineage across the European Alps.

Experimental design

The research question is more elaborated in this version. Some sections in methods can be omitted or shortened:
L450-454. This can be omitted
L532. I would replace regions by sites.
L546-554. This information seems to be redundant, as it was detailed in the paragraph above, please clarify.
L574-575. Since this section does not seem to include data from other studies in the Bavarian and Swabian Alps, I would change it to ‘..generated in this study’ and specify that it is only one site in each region.

Validity of the findings

The level of replication is low and the sampling is uneven. The section focusing on Sebacinales is however more interesting.
L256. .. across their natural range of distribution in the Bavarian Alps and the Swabian Alps. Only two sites were sampled, please re-write accordingly.

Additional comments

L221. Please add e.g. in your examples, as B. vivipara associates with ascomycetes like Cenococcum geophilum that do not belong to Pezizales.
L260. You can use other plant species here and elsewhere in the text.
L446. What are PCR profiles?
L641-645. There seem to be a low number of sequences per SH number, did you use the UNITE reference dataset (e.g. one sequence per SH)?
L641-649. This paragraph is a bit confusing, as results from this study and from the sequences retrieved from UNITE are intermixed. These could be written one after the other.
L709. What is a no SH match? A match at less than 99%?
L914-916. Please re-write for clarity.
L924. What are the regional levels?
L932. Please add reference/s. What are these trends?
Figure 1 is a bit misleading, as there are only two sites included in this study. Maybe they could be marked in the map with the other locations?

---

## Round 0.3 · accepted · Accept

I am pleased with the changes in your manuscript, I think it is ready now.